# Neural encoding with visual attention

**Meenakshi Khosla**[1,*]**, Gia H. Ngo**[1]**, Keith Jamison**[3]**, Amy Kuceyeski**[3,4] **and Mert R. Sabuncu**[1,2,3]

[1] School of Electrical and Computer Engineering, Cornell University, Ithaca, NY 14853
[2] Nancy E. and Peter C. Meinig School of Biomedical Engineering, Cornell University, Ithaca, NY 14853
[3] Radiology, Weill Cornell Medicine, New York, NY 10065
[4] Brain and Mind Research Institute, Weill Cornell Medicine, New York, NY 10065
[*] Correspondence: mk2299@cornell.edu

## Abstract

Visual perception is critically influenced by the focus of attention. Due to limited resources, it is well known that neural representations are biased in favor of attended locations. Using concurrent eye-tracking and functional Magnetic Resonance Imaging (fMRI) recordings from a large cohort of human subjects watching movies, we first demonstrate that leveraging gaze information, in the form of attentional masking, can significantly improve brain response prediction accuracy in a neural encoding model. Next, we propose a novel approach to neural encoding by including a trainable soft-attention module. Using our new approach, we demonstrate that it is possible to learn visual attention policies by end-to-end learning merely on fMRI response data, and without relying on any eye-tracking. Interestingly, we find that attention locations estimated by the model on independent data agree well with the corresponding eye fixation patterns, despite no explicit supervision to do so. Together, these findings suggest that attention modules can be instrumental in neural encoding models of visual stimuli. [1]

## 1   Introduction

Developing accurate population-wide neural encoding models that predict the evoked brain response directly from sensory stimuli has been an important goal in computational neuroscience. Modeling neural responses to naturalistic stimuli, in particular stimuli that reflect the complexity of real-world scenes (e.g., movies), offers significant promise to aid in understanding the human brain as it functions in everyday life [1]. Much of the recent success in predictive modeling of neural responses is driven by deep neural networks trained on tasks of behavioral relevance. For example, features extracted from deep neural networks trained on image or auditory recognition tasks are currently the best predictors of neural responses across visual and auditory brain regions, respectively [2, 3, 4]. While this success is promising, the unexplained variance is still large enough to prompt novel efforts in model development for this task. One aspect that is often overlooked in existing neural encoding models in vision is visual attention.

Natural scenes are highly complex and cluttered, typically containing a myriad of objects. What we perceive upon viewing complex, naturalistic stimuli depends significantly on where we direct our attention. It is well known that multiple objects in natural scenes compete for neural resources and attentional guidance helps to resolve the ensuing competition [5]. Due to the limited information processing capacity of the visual system, neural activity is biased in favor of the attended location [6, 7]. Hence, more salient objects tend to be more strongly and robustly represented in our brains. Further, several theories have postulated that higher regions of the visual stream encode increasingly

shift- and scale-invariant representations of attended objects after filtering out interference from surrounding clutter [8, 9]. These studies suggest that deployment of attention results in an information bottleneck, permitting only the most salient objects to be represented in the inferotemporal (IT) cortex, particularly the ventral visual stream which encodes object identity. These findings together indicate that visual attention mechanisms can be crucial to model neural responses of the higher visual system.

Visual attention and eye movements are tightly interlinked. Where we direct our gaze often quite accurately signals the focus of our attention [10]. This form of attention, known as overt spatial attention, can be directly measured by eye-tracking. Recent work has shown that fMRI activity can be used to directly predict fixation maps or eye movement patterns under free-viewing of natural scenes, suggesting a strong link between neural representations and eye movements [11]. In a similar vein, Sinz et al. [12] demonstrated that gaze shifts as estimated from pupil locations and behavioral states can be very useful in modeling spiking activity of mouse V1 neurons. More recent large-scale efforts in such concurrent data collection on humans, such as the Human Connectome Project (HCP) [13], that simultaneously record fMRI and eye-tracking measurements on a large population under free-viewing of movies, present a novel opportunity to probe the potential role of attention in neural encoding models of ecological stimuli.

Our contributions in this study are as follows:

- We demonstrate that leveraging information about attended locations in an input image can be helpful in predicting the evoked neural response. Particularly, we show that attentional masking of high-level stimulus representations based on human fixation maps can dramatically improve neural response prediction accuracy for naturalistic stimuli across large parts of the cortex.

- We show that it is possible to use supervision from neural response prediction solely to co-train a visual attention network. This training strategy thus encourages only those salient parts of the image to dominate the prediction of the neural response. We find that the neural encoding model with this trained attention module outperforms encoding models with no or fixed attention.

- Interestingly, we find that despite not being explicitly trained to predict fixations, the attention network within the neural encoding model compares favorably against saliency prediction models that aim to directly predict likely human fixation locations given an input image. This suggests that neural response prediction can be a powerful supervision signal for learning where humans attend in cluttered scenes with multiple objects. This signals a novel opportunity for utilizing functional brain recordings during free-viewing to understand visual attention.

## 2   Methods

Neural encoding models comprise two major components: a representation (feature extraction) module that extracts relevant representations from raw stimuli and a response model that predicts neural activation patterns from the feature space. We propose to integrate a trainable soft-attention module on top of the representation network to learn attention schemes that guide the prediction of whole-brain neural response. Our proposed methodology is illustrated in Figure 1.

**Feature extraction network**   We employ the state-of-the-art ResNet-50 [14] architecture pre-trained for object recognition on ImageNet [15] as the representation network to extract semantically rich features from raw input images. In this study, we focus on improving neural response prediction in higher-order regions of the visual pathway where receptive fields are larger and not limited to a single hemi-field. Prior evidence suggests that these regions are likely best modelled by deeper layers of object recognition networks [3, 16]. Thus, we extract the output of the last "residual block", namely res5 (after addition) before the global pooling operation to encode all images into a 2048-channel high-level feature representation image (of size $23 \times 32$, in our experiments), denoted as $F_{rep}$. All pre-trained weights are kept frozen during training of the neural encoding models.

**Attention network**   The attention network operates on the 2048-channel feature representation image $F_{rep}$. For simplicity, we employed a single convolutional layer that constructs the saliency map with a trainable $5 \times 5$ filter $V_{att} \in \mathbb{R}^{5 \times 5 \times 2048 \times 1}$ as, $S = G_\sigma * [V_{att} * F_{rep}]_+$. Here, $|\cdot|_+$ denotes

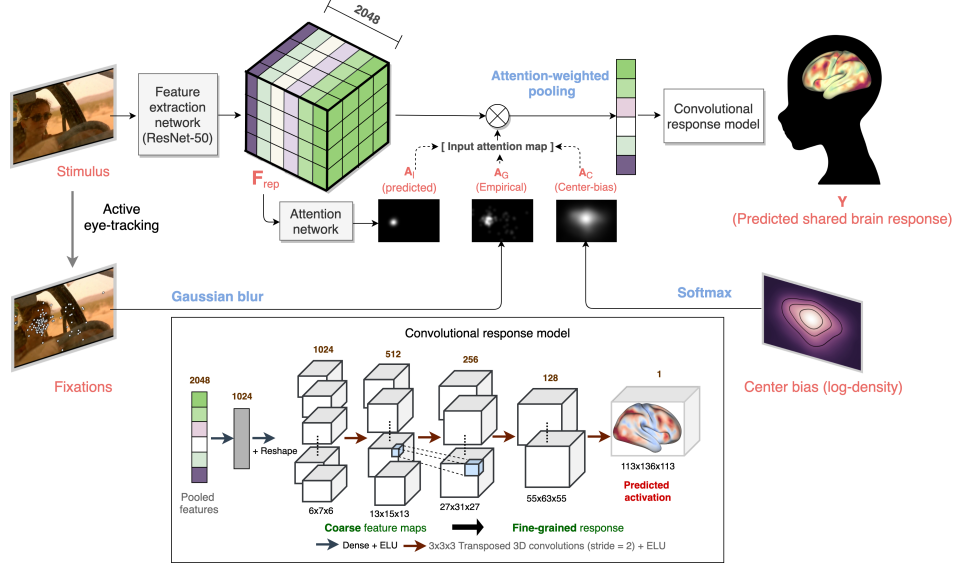

Figure 1: **Proposed method.** A trainable soft-attention module is implemented on top of a pre-trained representation network to rescale features based on their salience. The rescaled features are spatially pooled and fed into a convolutional response model to predict whole-brain neural response. We assess the value of the trained attention network by comparing it with neural encoding methods employing (i) stimulus-dependent attention maps derived from human fixations ($A_G$), (ii) stimulus-independent attention map derived from all fixations in the training set that reflects the center-weighted bias of our dataset ($A_C$) as well as a (iii) no attention model that spatially pools the features directly with no scaling.

the ReLU operation and $G_\sigma *$ indicates blurring using a $5 \times 5$ gaussian kernel with $\sigma = 1$. The attention scores for each pixel are finally computed from saliency maps by normalizing with the spatial softmax operation,

$$A_l^{(i)} = \frac{\exp S^{(i)}}{\sum_{j=1}^{n} \exp S^{(j)}}, i \in \{1, .., n\}. \tag{1}$$

Here, superscript $i$ is used to index the $23 \times 32$ spatial locations in the feature map $F_{rep}$. We note that existing literature on selective visual attention suggests a hierarchical winner-take-all mechanism for saliency computation, where only the particular subset of the input image that is attended is consequently represented in higher visual systems [7]. The softmax operation can be construed as approximating this winner-take-all mechanism. The attention is consequently applied as element-wise scaling to $F_{rep}$ to yield an attention modulated representation $F_{rep}^a = F_{rep} \odot A$.

**Convolutional response model**   The convolutional response model maps the spatially pooled attention modulated features $\mathbf{f}_g = \sum_{i=1}^{n} F_{rep}^{a(i)}$ to the neural representation space, reshapes them into coarse 3D feature maps and transforms them into an increasingly fine-grained volumetric neural activation pattern using trainable convolutions. This dramatically reduces the parameter count in comparison to linear response models with dense connections. Additionally, it captures spatial context and allows end-to-end optimization of the neural encoding model to predict high-resolution neural response, thereby alleviating the need for voxel sub-sampling or selection. The full sequence of feedforward computations in the convolutional response model are shown in the inset of Figure 1. The architecture of the convolutional response model is kept consistent across all CNN-based models to ensure a fair comparison.

## 2.1   Baselines and upper bounds

**No attention**   We compared the performance of all attention-based models against a model with no attention modulation that spatially pools the feature representation as, $\mathbf{f}_g = \sum_{i=1}^{n} F_{rep}^{(i)}$ (denoted as 'No

attention'). We implemented another baseline that uses the full feature map directly (instead of spatial pooling) as a flattened input to the convolutional response model. Due to computational/memory constraints, we had to reduce the dimensionality of the fully connected layer (to 256 units instead of 1024) in the convolutional response model for this encoding method. This model is henceforth denoted as 'No pooling'.

**Center-weighted attention** To further assess the usefulness of a learned attention network, we derive a stimulus-independent attention map ($A_C$) based on averaging across all eye gaze data in the training set, using Gaussian kernel density estimation. This essentially amounts to center-weighted attention (see Supplementary) since fixation locations on average are biased towards the center of an image [17]. The standard deviation of the Gaussian kernel is chosen to maximize log-likelihood on the validation set and is consequently set to 20.

**Gaze-weighted attention** We derive attention maps for every input frame from the eye gaze coordinates observed for the respective frame across different subjects. The human fixation maps are converted into attention maps $A_G$ by blurring with a Gaussian kernel of same standard deviation as the center-weighted attention model. The resulting attention maps in the original input image space are subsequently resized to the spatial dimensions of $F_{rep}$ and renormalized. Since these stimulus-specific attention maps are derived from actual human gaze information, they likely represent an upper bound in neural encoding performance among all attention-based models.

**Linear models** To date, neural encoding models in all prior work employ a linear response model with appropriate regularization on the regression weights. To compare against this dominant approach, we extract global average pooled (no-attention) features as well as pooled attention modulated features for both non-trainable attention schemes (center-weighted and gaze-weighted attention) as described above, to present to the linear regressor. We apply $l_2$ regularization on the regression coefficients and adjust the optimal strength of this penalty $\lambda$ through cross-validation using 10 log-spaced values in $\{1e-5, 1e5\}$. In later sections, we denote the performance of the above models as 'No attention (linear)', 'Center-weighted attention (linear)' and 'Gaze-weighted attention (linear)' respectively.

## 2.2 Training procedure

All parameters were optimized to minimize the *mean squared error* between the predicted and target fMRI response using Adam [18] for 25 epochs with a learning rate of 1e-4. Validation curves were monitored to ensure convergence and hyperparameters were optimized on the validation set.

## 2.3 Evaluation

**Neural encoding** We evaluated the performance of all encoding models on the test movie by computing the *Pearson's correlation coefficient* (R) between the predicted and measured fMRI response at each voxel. Since we are only interested in the stimulus-driven response, we isolate voxels that exhibit high inter-group correlations over all *training* movies. Inter-group correlation ("synchrony") values were computed by splitting the population into half and computing correlations between the mean response time-course of each group (comprising 79 subjects) at every voxel. We chose a data-driven metric (synchrony) to isolate the stimulus-driven cortex in order to avoid reliance on pre-defined atlases or functional localizers in identifying the voxels of interest. However, since choosing an arbitrary synchrony threshold may introduce a bias in the reported metrics, we employed a range of threshold values, from very loose (0.15) to very strict (0.75) for the correlation value to consider a voxel as "synchronous" [19]. Finally, to summarize the prediction accuracy across the stimulus-driven cortex, we compute the mean correlation coefficient across the synchronous cortex voxels by varying the "synchrony" thresholds from 0.15 (resulting in 160,900 voxels) to 0.75 (8,804 voxels). The spatial distribution of synchronous voxels across the brain as we vary the synchrony thresholds is illustrated in Figure 2(B). For region level analysis, ROIs were extracted using a population-wide multi-modal parcellation of the human cerebral cortex, namely the HCP MMP parcellation [20].

**Saliency prediction** Next, we wanted to assess if the learned attention model was indeed looking at meaningful locations in input images while predicting neural responses. To address this question and put the learned attention schemes in perspective, we assessed the agreement of predicted saliency

maps with human fixation maps for every frame in the test movie. Besides a qualitative evaluation, we computed quantitative metrics for comparing the predicted saliency maps against popular fixation (or saliency) prediction approaches. These include: (i) Itti-Koch [21]: a biologically plausible model of saliency computation that assigns pixel-level conspicuity values based on multi-scale low-level feature maps (intensity, color, orientation) computed via center-surround like operations similar to visual receptive fields, (ii) Deepgaze-II model [22]: a deep neural network based approach that extracts high-level features from a pre-trained image recognition architecture (VGG19) as input to a readout network that is subsequently trained to predict fixations using supervision from gaze data, and (iii) Intensity contrast features (ICF) model [22]: a low-level saliency computation model that uses the same readout architecture as the Deepgaze-II model, but on low-level intensity and intensity contrast feature maps as opposed to high-level features. Additionally, we also report evaluation metrics for the center-weighted saliency map. We note that the Deepgaze-II and ICF models were trained with eye-tracking supervision on the MIT1003 saliency dataset [23].

Developing metrics for saliency evaluation is an active area of research and several different metrics have been proposed that often exhibit discrepant behavior [24]. We report the most commonly used metrics in saliency evaluation [24], including, (i) Similarity or histogram intersection (SIM), (ii) Pearson's correlation coefficient (CC), (iii) Normalized scanpath saliency (NSS), (iv) Area under the ROC curve (AUC) and (v) Shuffled AUC (sAUC). Following [25], we used log-density predictions as saliency maps to compute all evaluation metrics.

## 2.4   Dataset

We study high-resolution 7T fMRI (TR = 1s, voxel size = 1.6  mm isotropic) recordings of 158 participants from the Human Connectome Project (HCP) movie-watching database while they viewed 4 audio-visual movies in separate runs [13, 26]. Each movie scan was about 15 minutes long, comprising multiple short clips from popular Hollywood movies and/or vimeo. Eye gaze locations of subjects were also recorded simultaneously at 1000Hz and resampled to 24Hz to match the video frame acquisition rate. All fMRI data was preprocessed using the HCP FIX denoising procedures, which include motion and distortion correction, high-pass filtering (2000 sec cut-off), head motion effect regression using Friston 24-parameter model (i.e., 6 rigid body motion parameters, their backward temporal derivatives and squares of those time series), automatic removal of artifactual timeseries by applying regression based on Independent Component Analysis (ICA) [27] as well as nonlinear registration to the MNI template space [28, 26]. Since the present study focuses on the development of population-wide predictive models, we averaged the response for each frame across subjects to obtain a single fMRI volume that represents the population average brain activation in response to that frame. After discarding rest periods as well as the first 10 seconds of every movie segment, we used about 12 minutes of audio-visual stimulation data per movie paired with the corresponding fMRI response and fixation data for analysis. We extract the last frame of every second of the video as a $720 \times 1280 \times 3$ RGB input to present as stimulus to the neural encoding models. The output is the predicted response across the entire brain, represented as a volumetric image of dimensions $113 \times 136 \times 113$. We estimate a hemodynamic delay of $4 \, sec$ using regression based encoding models (see Supplementary), as the response latency that yields highest encoding performance. Thus, all proposed and baseline models are trained to use the above stimuli to predict the fMRI response 4 seconds *after* the corresponding stimulus presentation. We train and validate our models on three movies using a  9:1 train-val split and leave the fourth movie for independent testing. This yields 2000 training, 265 validation and 699 test stimulus-response pairs.

## 3   Results

**Incorporating gaze-weighted attention significantly improves neural response prediction.**   We first examined whether attention weighted pooling helps to improve response predictions. Figure 2 shows the mean prediction accuracy across the entire synchronous cortex for all models considered in this study. We find that the 'gaze-weighted attention' model significantly outperforms the 'no attention' model for both linear ($\sim 40$ % improvement among all voxels with synchrony>0.15), as well as convolutional response model ($\sim 47$ % improvement among all voxels with synchrony>0.15). The attention maps result in amplification of features of attended locations along with suppression of other irrelevant information. This re-scaling of features before pooling using fixation patterns obtained from eye-tracking data remarkably improves neural encoding performance across large areas of the

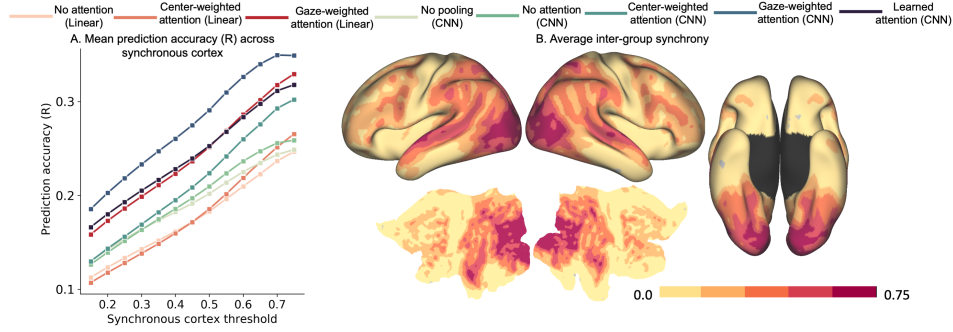

Figure 2: **Quantitative evaluation of all models.** (A) depicts mean correlation values across the synchronous, (i.e., stimulus-driven) cortex defined at a range of synchrony thresholds ([0.15,0.75]). Each point thus reflects the mean prediction accuracy for a model across all voxels within synchronous cortex defined by a threshold value (x-axis). (B) depicts the inter-group correlation (synchrony) values across the entire human cerebral cortex.

cortex, suggesting that neural responses are indeed dominated by sensory signals at attended locations. Although we employed a convolutional response model primarily for computational efficiency in predicting a high-resolution (113x136x113) whole-brain neural response, we also observed a small improvement in neural encoding with this response model in comparison to a linear response model.

**Trainable attention model outperforms models with no attention or center-weighted attention** In addition to improving neural response prediction, the convolutional response model renders end-to-end training of encoding models on whole-brain neural data feasible by dramatically reducing the number of free parameters in comparison to linear response models. In this study, we exploited this increased parameter efficiency to co-train an attention network on top of a pre-trained representation network (while freezing the representation network) for the goal of neural response prediction. As shown in Figure 2, the encoding model with learned attention surpasses models with no pooling, no attention or center-weighted attention in mean prediction accuracy across the sychronous cortex as well across most ROIs involved in object processing. This suggests that even with no eye-tracking data, as is the case with majority movie-watching fMRI datasets, modelling visual attention as re-weighting of stimulus representations based on spatial attention masks can still be beneficial in response prediction. The improvements are most apparent in ventral stream regions such as the Fusiform Face Complex (FFC) and PIT Complex, as well as object-selective parts of the lateral occipital complex (LO1, LO2, LO3) (Figure 2). Studies in visual perception have shown that these lateral occipital areas respond more strongly to intact objects than scrambled objects or textures, providing strong evidence for their role in object recognition as well as object shape perception [29, 30, 31]. Accuracy in another group of areas within the temporo-parieto-occipital junction, which is known to be involved in visual object recognition as well as representation of facial attributes such as the intensity of facial expressions [32], is similarly improved with the trained attention network. In addition to these areas, we also observe some improvement in neural encoding performance in other higher order processing regions across the dorsal visual stream, motion-sensitive visual regions (MT+ complex) and their neighboring visual areas (Figure 3). We also trained the proposed and baseline models on representations of other randomly selected deep layers within the ResNet-50 architecture and observed a similar benefit of attention modulation across different layers (see Supplementary). Further, a representational similarity analysis comparing non-modulated and attention modulated representations of different layers across popular architectures showed that models that explain stimulus-dependent human fixation patterns are able to better account for the representational geometry of neural responses across intermediate and higher visual object processing areas (see Supplementary). Taken together, these findings provide further support for the utility of attention modelling in neural encoding approaches. In addition to improving accuracy, the attention model further affords interpretability by highlighting salient locations within the input image that are being employed to make response predictions.

**Learned attention policies correspond remarkably well with human fixation maps.** Figure 4 depicts saliency maps predicted by the trained attention network on sampled frames from the test

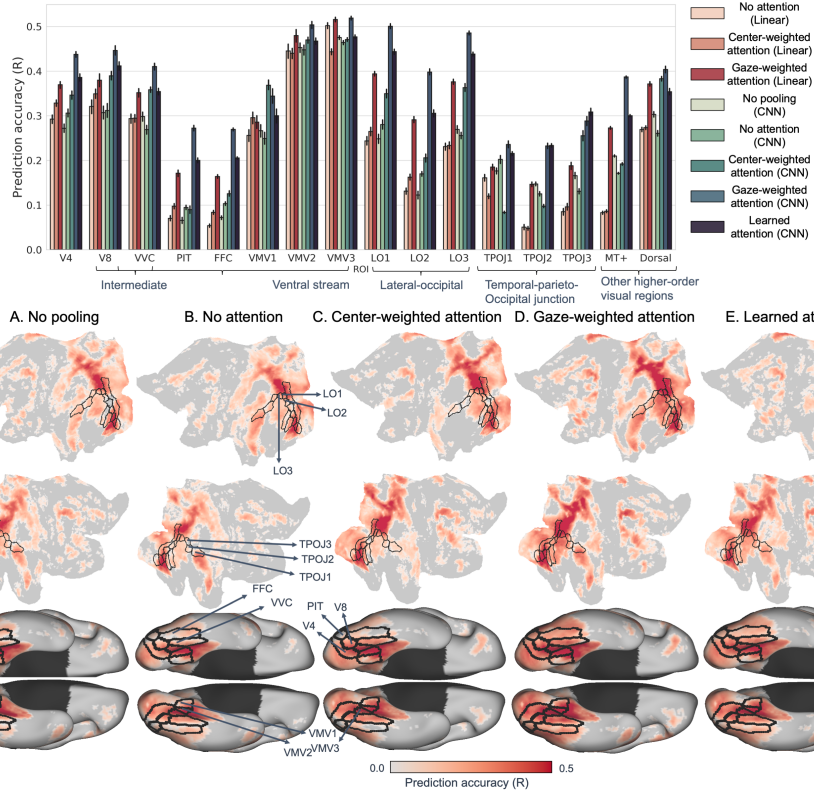

Figure 3: **Top: ROI-level analysis** Mean correlation values across intermediate (V4), higher visual areas in the inferotemporal cortex and its neighborhood and other higher higher-level visual regions (Dorsal, MT+) as described in the HCP MMP parcellation [20]. Error bars represent 95% confidence intervals around mean estimates computed using bootstrap sampling. **(A)-(E) Prediction accuracy across the cortical surface for all deep CNN-based models.** Statistical significance of individual voxel predictions is computed as the p-value of the obtained sample correlation coefficient for the null hypothesis of uncorrelatedness (i.e., true correlation coefficient is zero) under the assumptions of a bivariate normal distribution. Only significantly predicted voxels (p<0.05, FDR corrected) for each method are colored on the surface. Prediction accuracy maps for encoding methods with linear response models are provided in the Supplementary.

movie. This qualitative assessment indicates that the proposed neural encoding model learns attention policies that are consistent with human fixation maps. Since attention is learned on top of high-level features, the model learns to focus on high-level stimuli features such as the presence of faces, hands and more conspicuous objects likely to direct attention in natural scenes. A closer look at incongruent cases indicates that images where the model fails to track human fixations are often highly complex scenes, where fixations may be driven by contextual knowledge of previous movie frames (Figure 4, top-right) or auditory signals, e.g., who the speaker is, etc. (Figure 4, bottom-right).

Table 1 shows quantitative metrics that compare the quality of saliency maps computed by benchmark models trained to predict gaze on our data. We also listed the performance of the attention network that was merely trained on fMRI data, and not eye gaze data. We note that our attention network performs on par with popular fixation prediction models that are trained directly on the task of saliency prediction in a supervised manner (ICF and Deepgaze-II). This trend holds for almost all saliency evaluation metrics, as shown in Table 1. This observation is particularly interesting given that the attention network is trained using supervision from neural response prediction only, without any information about gaze coordinates.

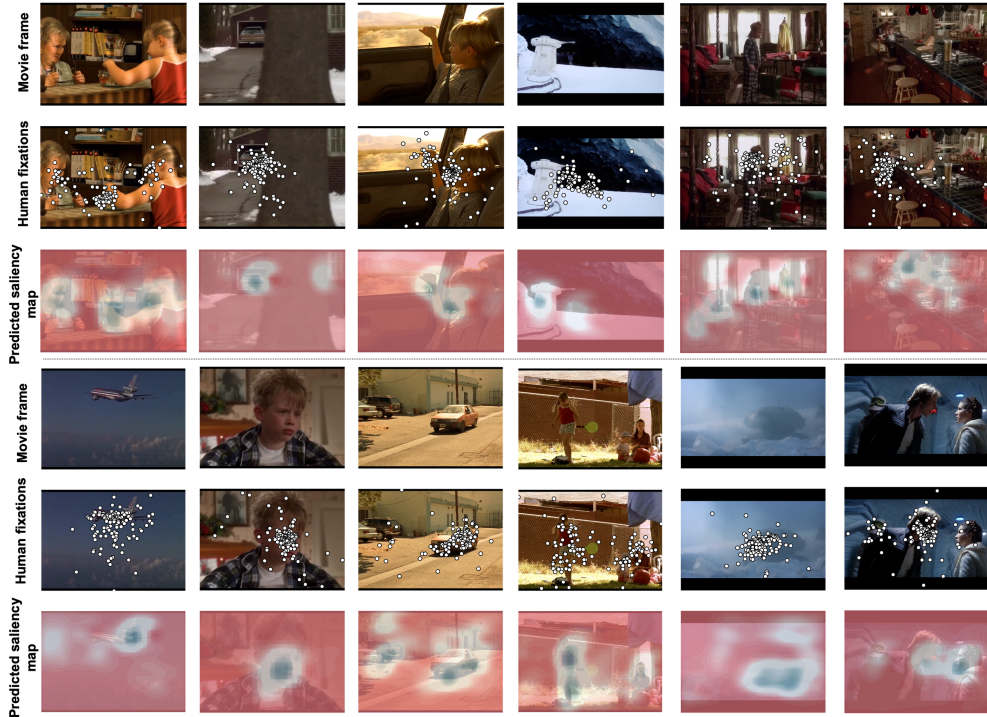

Figure 4: **Qualitative assessment of saliency (log-density) maps.** Top row shows sampled frames from the test movie, middle row shows human fixation maps overlaid on the corresponding frame, bottom row shows saliency maps predicted by the attention network of the proposed neural encoding model. Blue indicates high saliency values whereas red indicates low saliency.

Table 1: Evaluation against saliency prediction models. Mean and standard errors for each metric are reported. Best results are bolded.

| Model | SIM ↑ | CC ↑ | NSS ↑ | AUC ↑ | sAUC ↑ |
|---|---|---|---|---|---|
| Itti-Koch | $0.318 \pm 0.002$ | $0.325 \pm 0.004$ | $1.010 \pm 0.014$ | $0.795 \pm 0.004$ | $0.537 \pm 0.006$ |
| ICF | $0.291 \pm 0.002$ | $0.190 \pm 0.007$ | $0.646 \pm 0.023$ | $0.665 \pm 0.006$ | $0.647 \pm 0.005$ |
| Center-weighted | $0.327 \pm 0.002$ | $0.350 \pm 0.004$ | $1.074 \pm 0.013$ | $0.803 \pm 0.003$ | $0.496 \pm 0.006$ |
| Deepgaze-II | $0.359 \pm 0.003$ | $\mathbf{0.420} \pm 0.005$ | $\mathbf{1.425} \pm 0.025$ | $\mathbf{0.808} \pm 0.004$ | $\mathbf{0.713} \pm 0.004$ |
| Ours | $\mathbf{0.392} \pm 0.004$ | $0.403 \pm 0.010$ | $1.375 \pm 0.041$ | $0.754 \pm 0.006$ | $0.645 \pm 0.006$ |

## 4 Discussion and Conclusion

In the present study, we demonstrate that encoding models with visual attention, whether explicitly estimated from human fixation maps or modelled using a trainable soft-attention scheme, yield significant improvements in neural response prediction accuracy over non-attention based counterparts. We observe consistent improvements across most high-level visual processing regions, suggesting that unattended portions of an input image may likely have little effect on neural representations in these regions. Loosely, this aligns well with Treisman's feature integration theory [33], which proposes that integrated object representations are only formed for attended locations. In addition to improving response prediction accuracy, inclusion of visual attention within neural encoding models promises a better understanding of spatial selection and its influence on neural representations and perceptual processing. Further, while our study integrates a spatial attention module within a neural encoding model, the proposed approach is not restricted to this particular form of attention. For example, spatially global feature-based attention can also be studied within the context of neural encoding models as "channel-wise" attention-weights instead of spatial attention masks. We believe the observation that neural response prediction may be a useful supervision goal to study attentional deployment is particularly exciting and can be extended in novel ways.

The saliency of a stimulus often depends on the context within which it is presented and attentional selection strategies can be modulated by task demands [8]. Importantly, attention selects across space and time; here, we focus on spatial selection of stimuli but it is likely that modeling temporal context can lead to substantive improvements. Context can also help in highlighting attentional targets that may be driven by "surprise". Thus, in movie watching, future neural encoding models should also capture the sequence of frames, rather than isolated frames, and the audio track in modeling attention.

Our study provides a first attempt in capturing visual attention within neural encoding models. We see several opportunities for extending this work. In the present framework, we employed attention as a masking strategy to filter out clutter and retain information from only the most relevant (i.e. attended) parts of an image. It would be interesting to study how and where the features of ignored stimuli (i.e. the stimuli that doesn't get past the attentional bottleneck) are encoded. Further, here, we modeled attention on top of high-level semantic features. In principle, the attention network can be implemented on top of any level within the representation network hierarchy, including lower stages and understanding where attention computations leads to best neural prediction accuracy and/or agreement with human fixation maps could be a worthwhile exploration. A straightforward extension in this direction would be to add the attention module on top of both low-level cues and high-level representations or to combine feature maps across layers before presenting to the attention network. In the future, we aim to further explore novel ways of incorporating attention within neural encoding models.

Beyond advancing our understanding of sensory perception, neural encoding models have potential for real-world applications, most obviously for brain-machine interface. Additionally, an improved understanding of the link between sensory stimuli and evoked neural activation patterns can provide opportunities for neural population control, for e.g., by synthetically designing stimuli to elicit a specific neural activation pattern [34].

## Broader Impact

Understanding the link between sensory stimulation and evoked neural activity in humans as revealed with encoding models, can provide foundations for developing novel therapies. Viewed in this regard, an improved understanding of information processing in the brain has tremendous potential. However, encoding models can be very sensitive to biases in the training set. Our models were trained using data from the Human Connectome Project database. While this large-scale project has made a lot of valuable data publicly available to the scientific community for studying brain structure and function, it is important to consider the representational bias in the dataset. For instance, the data we analyzed is exclusively limited to a young adult population. Such biases can possibly lead to poorer generalization of models trained with these large-scale datasets on other population groups that are inadequately represented. Once these encoding models are ripe for therapeutic applications, this dataset bias could prevent under-represented groups from deriving the benefits of a useful technology, resulting in uneven access across populations. Given these considerations, it is important to address potential representation biases in fMRI datasets and develop solutions for improving diversity and inclusion. More generally, fMRI studies involving human subjects can raise a wide range of other ethical issues as well, including data privacy issues and informed consent.

Further, one should be cautious about the deployment of attention or gaze prediction models in applications such as advertising. Given the value of eye tracking based attention in marketing spaces, public policy notices or political campaigns, it is important to be wary of a malicious use of these attention prediction methods for profit-seeking or by ill-intentioned parties seeking to further their own agendas. These applications regard attention as a commodity to be captured and the adopted technologies can be used to manipulate users in subtle ways. An improved understanding about the link between stimuli and perceptual processing in the brain, as provided with encoding models, can also be exploited to further design or identify stimuli likely to elicit a specific emotional or cognitive response. The fact that these technologies can be deployed without the targeted individual's knowledge or consent indicates it is important to protect users from the vulnerabilities exploited by these agents.

## Acknowledgments and Disclosure of Funding

This work was supported by NIH grants R01LM012719 (MS), R01AG053949 (MS), R21NS10463401 (AK), R01NS10264601A1 (AK), the NSF NeuroNex grant 1707312 (MS), the NSF CAREER 1748377 grant (MS) and Anna-Maria and Stephen Kellen Foundation Junior Faculty Fellowship (AK).

## Footnotes

[1]Our code is available at `https://github.com/mk2299/encoding_attention`.

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
