[Supplementary Material]

# Supplementary Information
# Neural encoding with visual attention

**Meenakshi Khosla[1,*], Gia H. Ngo[1], Keith Jamison[3], Amy Kuceyeski[3,4] and Mert R. Sabuncu[1,2,3]**

[1] School of Electrical and Computer Engineering, Cornell University, Ithaca, NY 14853
[2] Nancy E. and Peter C. Meinig School of Biomedical Engineering, Cornell University, Ithaca, NY 14853
[3] Radiology, Weill Cornell Medicine, New York, NY 10065
[4] Brain and Mind Research Institute, Weill Cornell Medicine, New York, NY 10065
[*] Correspondence: mk2299@cornell.edu

## Model comparison across randomly selected layers

Here, we wanted to examine if the learned attention model would lead to performance improvements in neural response prediction across other deep layers as well. We trained all 8 models using stimuli representations $F_{rep}$ from 2 randomly selected layers in the res5 block of the pre-trained ResNet-50 architecture, namely 'add_14' and 'res5c_branch2b'[1], henceforth denoted as 'Random ResNet-50 layer 1' and 'Random ResNet-50 layer 2' respectively. Figure 1 shows the prediction accuracy across the synchronous cortex on the held-out movie for all models. We again observe that the learned attention model performs favorably against models with no attention, no pooling or center-weighted attention. Further, the gaze-weighted attention method outperforms all other methods employing the same response model (linear or convolutional), consistent with our previous findings.

## Representational similarity analysis

Representational similarity analysis (RSA) is a popular framework to compare representations of a computational model against cortical representations [1, 2]. It can be used to directly measure a computational model's ability to explain the representational geometry in neuronal responses. Here, we wanted to assess the impact of attention modulation on a computational model's alignment to brain responses for a wider range of model layers and architectures. Given stimuli from the held-out movie (699 frames) and the corresponding response (after hemodynamic lag), we implemented the following procedure for time-continuous RSA: (i) We computed Pearson's correlation distance (1-R) between the response vectors for every pair of test frames to obtain the representational dissimilarity matrix (RDM) of neural responses. The dissimilarity matrices are averaged across subjects to yield a population-averaged 'neural' RDM. The region of interest (ROI) mask for extracting response vectors to estimate neural RDMs was derived from all voxels in intermediate (V4), ventral visual stream and lateral occipital ROIs. Responses of all voxels were normalized using z-scores before computing the dissimilarity matrix. (ii) We extracted model representations from intermediate layers of 3 pre-trained (ImageNet) architectures, namely ResNet-50 (res2, res3, res4, res5), VGG-16 (maxpool1, maxpool2, maxpool3, maxpool4, maxpool5) and AlexNet (conv1, conv2, conv3, conv4, conv5). For each of these representations, we further computed attention modulated representations using attention maps computed with each saliency prediction method as described above. For the Itti-Koch model, we used normalized saliency as the attention map. For all remaining saliency models, we used probabilistic density predictions as attention maps. All attention maps were resized to the spatial dimensions of the respective layer for this computation. Representational vectors were compared pair-wise in terms

Figure 1: **Quantitative evaluation.** Mean correlation values across the synchronous, (i.e., stimulus-driven) cortex defined at a range of synchrony thresholds ([0.15,0.75]). Each point thus reflects the mean prediction accuracy for a model across all voxels within synchronous cortex defined by a threshold value (x-axis).

Figure 2: **Representational similarity analysis(RSA).** y-axis measures the agreement between 'model' RDMs and 'neural' RDMs based on their rank correlation measure. x-axis is use to index the layer (index 1 refers to the earliest layer of the architecture) and the saliency method used for attention masking of the features before pooling.

of their Pearson correlation distance (1-R) to obtain the 'model' RDM. (iii) Finally, we compared the compatibility of the neural and model RDMs by using a rank correlation measure (Kendall's $\tau_A$).

As shown in Figure 2, prioritized selection of stimulus features based on saliency significantly improves the correlation of model RDMs with neural RDMs. This trend holds for most models and layers, suggesting that the benefits of attentional masking are not restricted to forward encoding models alone, but may be more universal. Further, we find that models that better explain stimulus-dependent human fixation patterns (such as Deepgaze-II or the learned attention model) are able to better account for the representational geometry of neural responses across higher visual object processing areas.

## Regions of interest (ROI)

We employed the HCP MMP parcellation for all ROI-level analysis. Dorsal and ventral visual stream ROIs as well as MT+ ROIs in Figure 3 (main text) were derived from the explicit stream segregation

Figure 3: **A. Center-weighted saliency map** and **B. Eye tracking statistics**

and categorization described in the HCP MMP parcellation [3] and are defined here in Table 1 for quick reference.

Table 1: ROI categorization

| Group | ROIs |
|---|---|
| Dorsal | V3A, V3B, V6, V6A, V7, IPS1 |
| Ventral | V8, VVC, PIT, FFC, VMV1-3 |
| MT+ | MT, MST, V4t, FST |
| Lateral occipital | LO1, LO2, LO3 |

## Center-weighted attention

Figure 3 depicts the center-weighted saliency map used in all center-weighted attention models. We also report per-movie eye tracking statistics therein from all frames used for training or testing the models. We note that not all subjects had eye tracking measurements for every frame in the movies. Figure 3B shows the number of subjects for which eyetracking data was available per movie (distribution across frames). This suggests that despite the missing data, most frames among all training and testing movies (MOVIE 4) had recorded gaze coordinate measurements from ∼110-130 subjects.

## Voxel-wise prediction accuracy (R) of linear models

Figure 4 depicts the prediction accuracy across the cortical surface for all methods employing linear response models that were considered in this study. As can be seen clearly, just as in methods with CNN response models, gaze-weighted attention significantly improves prediction accuracy across most higher order visual areas over models with no attention or center-weighted attention.

## Estimating hemodynamic (BOLD) response delay

fMRI BOLD response delay was estimated using the baseline 'No attention (Linear)' encoding model due to its computational efficiency in comparison to encoding models employing convolutional response models. The input to these models was the 2048 dimensional (average pooled) representation of the stimuli, and the output was the evoked fMRI response across the synchronous cortex (i.e., voxels with synchrony>0.15) at different lags (1-7 seconds) from the stimulus. Thus, the output is a 160900-D vector corresponding to the fMRI response. All models were trained with 5-fold cross-validation using the stimulus-response pairs from the *training* dataset only.

Based on Figure 5, we estimated a response delay of 4 seconds, as this lag consistently yielded the maximum prediction accuracy across 5-fold cross validation. Thus, all encoding models described in the main text were trained to predict fMRI response *after* 4 seconds of stimulus presentation.

Figure 4: **Prediction accuracy across the cortical surface for all methods using linear response models.** Statistical significance of individual voxel predictions is computed as the p-value of the obtained sample correlation coefficient for the null hypothesis of uncorrelatedness (i.e., true correlation coefficient is zero) under the assumptions of a bivariate normal distribution. Only significantly predicted voxels ($p<0.05$, FDR corrected) for each method are colored on the surface.

Figure 5: **Hemodynamic response delay.** 5-fold cross-validated prediction accuracy (R) of the simple ('No attention') model on the training dataset. Error margins are computed from the standard deviation of prediction accuracy across the 5 folds.

# Predicted saliency maps for the entire held-out movie

The following figures show the fixation maps and corresponding saliency maps predicted by the attention network of the proposed neural encoding model for frames sampled every 4 seconds from the held-out movie.

## Footnotes

[1]Notation from pre-trained ResNet-50 model: https://keras.io/api/applications/resnet/