[Reviews · NeurIPS 2020]

Review 1

Summary and Contributions: By adding a soft attention module, the authors propose a novel neural encoding model of (overt bottom-up) visual attention. The results are convincing even though at times the findings are more or less known (from neuroscience literature), the description of experiments would benefit from further details (both the computational experiments, and the fMRI acquisition where it is particularly weak), and some evaluation decisions may be easily criticized. The model clearly addresses one specific type of attention. It is unclear how an analogous model will be generated for other types of attention, or even more interestingly, whether we can find a somewhat more general-purpose model for the many types of attention. This is not a demerit of the paper; instead is my way of saying that this paper gave me food for thought! Overall, very good work.

Strengths: + The simplicity of the idea accompanied by convincing results, some with such high degree of nomological validity that were highly expected. + Relevance of the topic and timeliness. + Clear and explicitly mentioned contributions.

Weaknesses: + Description of methods. + Sometimes the findings were already known. While good for validity, but they can hardly be called findings.

Correctness: Despite the limited description, but they are correct as far as I can tell. Even though I cannot know the specifics of many decisions and/or the parameterization, in the overall picture, should have I been conducting the research, I would have taken very similar experimental decisions.

Clarity: Very much! Even though I feel the description of the methods could be given more detail, but the overall reading is smooth and easy to follow.

Relation to Prior Work: This is elegantly spread throughout the paper. There is, of course, the common early presentation of the state of the art in the introduction, but other important references are given in due time as the paper progresses. Nomological validity is attempted, but here, I perhaps feel that more could have been said. The topic is interesting. Until last year, I would have agreed that the topic had been overlooked, but suddenly, in the last year only I have already seen 3 papers on the topic by different groups. These are not cited, possibly suggesting that being so recent, the authors are indeed not aware of these. But at the same time confirming that the topic is becoming hot very recently. In this sense, I feel this paper is timely!

Reproducibility: Yes

Additional Feedback: Major + Surely, if you learn a "whole brain model" the purpose of the trainable soft-attention module should somewhat be captured by the ventral network [CorbettaM2008], and in fact, you mention later than the ventral stream is more prominent. Why then is this additional module needed? While I can see its effectiveness from a computational point of view, but it looks biologically unplausible. This is not ideal from a computational neuroscience point of view. Can the authors give a little bit more insight on the rationale of this module? + Allow me the acidity in this comment, but I am not aware of such thing as a “standard pipeline” for fMRI acquisition and processing. Please provide the details (in the supplementary material if you want) and do so in some de facto standard manner e.g. [PoldrackRA2008] or other. + Regarding the finding that learned attention policies correspond remarkably well with human fixation maps, what would happen on covert attention or top-down processes? + Regarding the finding that neural responses are [indeed] dominated by sensory signals at attended locations; well this is true as far as I know for bottom-up attention, but this is already known. What’s the novelty here? + When evaluating the prediction against the observed fMRI you opt to pick some voxels with high inter-group correlations. But isolating the voxels with high inter-group correlations will naturally drive/bias the subsequent estimation of the Pearson correlation coefficient between model predictions and fMRI observations. That is analogous to say, I correlate with the voxels that I previously chose. In my opinion, that is a major flaw in the validation process. While it does not invalidate the findings, but it means that they should be interpreted with care. Minor + The last of the contributions claims that “despite not being explicitly trained to predict fixations, the 60 attention network within the neural encoding model compares favorably against saliency 61 prediction models”. Even though not trained to predict fixations, but fixations are used during training! That is, the inverse model would anyways be informed of them. While still a valid observation but perhaps not that unexpected. + The statement that “more salient objects tend to be more strongly and robustly represented in our brain” is as far as I know, only true for bottom-up overt attention. If this statement is intended to be more general, can the authors add a reference? If it is intended to be specific, then some rephrasing may be convenient. + The statement that “modelling visual attention can [still] be beneficial in response prediction” appears to be obvious. Why would not modelling be better than modelling? =========== After Rebuttal =========== Congratulations to the authors! After seeing the rebuttal, not only I confirmed the high score but further I joined the recommendation for oral.


Review 2

Summary and Contributions: The authors used attention to improve prediction of voxel activation in fmri experiments. They then used their attention mechanism to predict fixations in human subject, and achieved near state-of-the art performance. Update post-author-response: thanks for your reply; no changes in score.

Strengths: For me, the main strength of the paper is that they could train their attention mechanism on fmri data and use that to predict fixations. In hindsight not massively surprisingly, but still very cool.

Weaknesses: No major weaknesses. Other than that I'm not sure what, ultimately, it tells us about the brain, as the results are not especially surprising. However, I don't do fmri (or train networks), so I would say that's a minor weakness at worst.

Correctness: I think so.

Clarity: I'm not a big fan of the organization: you had to remember tons of things before getting to the point. But after reading it three times I more or less figured out what was going on.

Relation to Prior Work: Not an expert, so I can't really say.

Reproducibility: Yes

Additional Feedback: None.


Review 3

Summary and Contributions: The study shows that incorporating gaze information (human fixation maps) into a neural "encoding" model significantly improves fMRI response predictions. A second major advance is that the authors train a visual attention network to accurately predict neural response using on "salient" regions of images, without actually using gaze data. The predicted saliency maps from the attention network correlate well with other saliency prediction methods and show high correspondence with the actual gaze maps.

Strengths: I would rate the study highly on multiple fronts including the significance and novelty of the contribution. As a researcher working at the interface of deep learning and functional MRI, I think the findings are timely and of high relevance to the NeurIPS community. Additional strengths: * They show, using a rather parsimonious attention model, evidence for improvement in neural response prediction accuracies. * Apt controls and comparisons employed throughout (e.g. center-weighted attention, DeepGazeII)

Weaknesses: I would not call the following major weaknesses, yet the study would benefit from paying attention to these points: * As a part behavioral neuroscientist by training, I would be careful with the use of the words attention vs. gaze, which the authors use interchangeably. Psychophysicists distinguish attention from gaze, and the focus of gaze is not always the focus of attention. For example, I can fixate at one point in space while covertly attending to another, distal object. * It would help to see if the model could also capture inter-subject variations in saliency maps, rather than simply subject-averaged predictions. * Some formal statistical tests of performance improvement could have been reported, although I do realize this is not conventional for conference fora. * Softmax operation is somewhat different from ‘winner-take- all’. Whether attention is a hard spotlight or a soft focus is debated (line 90).  * The trend of increase in prediction accuracy with increase in synchronous cortex threshold in fig 2 is expected since with greater inter-group synchrony, a population-wide model is expected to do better. It could have been interesting to note the number of voxels with varying synchronous cortex thresholds (fig 2). * Given the kind of noise in fMRI data, the numbers reported vis-a-vis saliency prediction are commendable (Table 1). Nonetheless, it would be good to comment on what can be done to actually match/surpass state of the art (DeepGazeII), as the proposed model does fall short of it.

Correctness: To my best knowledge, all claims and methods appear accurate.

Clarity: The paper is well structured. It is easy to read and follow the logic of the arguments.

Relation to Prior Work: Despite an elaborate Introduction, the relationship to previous work is not very clearly addressed in the paper. Nevertheless, this is such a new field that I doubt that there are too many similar papers in the community. I feel the results are sufficiently novel.

Reproducibility: Yes

Additional Feedback: I appreciate the efforts by the authors to address my comments. My score remains unchanged.


Review 4

Summary and Contributions: This paper packs a lot into 8 pages. It is unfortunate that I like to print out the paper to review it (avoids internet distractions) and many authors now rely on readers using a pdf viewer to embiggen the figures. I was unable to read some figure details until I looked at it on my laptop. Anyway, the point of this paper is that using eye tracking data to use as an attentional filter on the last convolutional layer of resnet 50 greatly improves neural response (fMRI) prediction. This is a nice result. Another importanct point is using a (shallow) convolutional network to make the predictions, which reduces the parameters a lot. The system greatly outperforms previous simple linear predictors. The second main result is that training an attentional filter directly while predicting fMRI response leads to a salience map that is competitive with the best supervised salience models (e.g., Deepgaze-II. This is a remarkable result, given that no eye-trackjing daeta was used. Of course, backprop will always focus in on the most predictive inputs, so perhaps this should have been expected, but this result is especially cool because it means we can now use this approach on fMRI datasets for which eye tracking data is not available. I have read the authors' response (which responded to my very minor criticism) and am convinced that this paper should be accepted.

Strengths: 1. The paper proposes a novel method of predicting neural coding by augmenting the network with eye movement data, enhancing the regions of the image that are being foveated by the viewers (on average). 2. The paper proposes a simple convolutional network instead of the usual linear mapping from a deep network representation to the neural representation (fMRI responses). 3. The method gives much better prediction results than previous methods. 4. They then show that just letting the network learn an attentional filter also improves the results, although not quite as much as using the eye movement data. 5. The attentional filter, which is only trained to predict the neural response, gives salience maps that are competitive with state of the art salience maps trained by supervised learning from eye movement data. 6. Given this, they propose that the same method (training the network to learn its own attentional filter) will work well with fMRI data that does not have accompanying eye movement data.

Weaknesses: The scale of the heatmaps in Figures 4 and the supplementary salience figure should be provided.

Correctness: It appears correct to me.

Clarity: Yes, very clear.

Relation to Prior Work: Yes. However, while I am not aware of similar work, there could be some work that I missed. I hope if there is relevant work, I hope the other reviewers will know about it.

Reproducibility: No

Additional Feedback: I would expect that the authors will provide the code for this work. Otherwise, it could be difficult to reproduce without significant effort.

[Author Response · NeurIPS 2020]

We thank all reviewers for their insightful feedback and endorsements. We will incorporate all suggestions in our final version and have addressed the main comments and questions below:

**[R1 - Biological plausibility and rationale behind the soft-attention module]** The proposed attention module is inspired by a core property of information processing in the brain, namely, its flexible and selective nature in the face of limited neural resources. By incorporating the module within the encoding network, we can model this selective process to select certain portions of visual stimuli (the attention "spotlights") for subsequent processing at the expense of others. The use of multiplicative scaling factors to modulate feature maps has some biological grounding insofar as it is loosely inspired from the notion of gain modulation in existing studies on biological attention. A biologically plausible computational model of attention would capture both bottom-up as well as top-down influences of working memory and context as they may ultimately constrain which locations are selected. While this is beyond the scope of our present work, we believe the findings presented here provide further motivation for future work in this direction.

**[R1 - "standard" fMRI preprocessing]** We fully agree. We will report the data pre-processing operations following the guidelines recommended in the suggested study.

**[R1 - Evaluation]** We agree with the limitations. We wanted to isolate the stimulus-driven cortex in a data-driven manner rather than relying on pre-defined atlases or other task-based functional localizers to identify voxels of interest. We further identified these voxels solely based on the inter-group correlations within the training dataset and presented the performance of different models at varying thresholds of synchrony from very loose (0.15) to very strict (0.75). However, we do understand that this evaluation methodology may still induce biases and we will acknowledge the limitations in the discussion. The number of voxels varies from 160,900 to 8,804 as we vary the synchrony threshold from 0.15 to 0.75. We will also include this trend in our final version.

**[R1 - fixations during training and the last contribution]** Perhaps we were not clear in the statement. We here refer to the learned attention model that does not employ fixation data during training or testing. The attention network is trained on top of the representation network for the goal of neural response prediction. As a consequence, this network only requires stimulus-response pairs and no eye-tracking data.

**[R1 - modeling v/s not modeling]** We will rephrase the statement to be more specific. Here, we mean that modeling attention as re-weighting of stimulus representations based on spatial attention masks is beneficial in response prediction.

**[R3 - Attention vs gaze]** We agree with the reviewer's point. This is an important distinction and we will add the clarification in the introduction section to resolve this difference and shed light on the different types of attention.

**[R3 - Inter-subject variations in saliency maps]** This is an intriguing direction and is part of our ongoing work.

**[R3 - Future improvements for gaze prediction]** There are several ways in which saliency prediction can be improved with our method. Here, we focused on simplicity as a proof of concept. A straightforward extension would be to add the attention module on top of both low-level cues and high-level representations or to combine feature maps across layers before presenting to the attention network. Further, attention selects across space and time - here we focus on spatial selection of stimuli but it is likely that modeling temporal context can lead to substantive improvements. Context can also help in highlighting attentional targets that may be driven by "surprise". We will expand on this in the Discussion.

**[R1 - One particular type of attention]** Our study integrates a *spatial attention* module within a neural encoding model. However, the proposed approach is not restricted to this particular form of attention. For example, spatially global *feature-based attention* can also be studied within the context of neural encoding models as "channel-wise" attention-weights instead of spatial attention masks. We believe the observation that neural response prediction may be a useful supervision goal to study attentional deployment is particularly exciting and can be extended in novel ways. Promisingly, as pointed out by **R4**, our study highlights that attention can be studied even with the majority of naturalistic fMRI datasets with no eye tracking.

**[R1 - Relationship to prior work]** We apologize for missing any related prior work. Upon further literature search, we came across a somewhat related paper modeling mouse V1 spiking activity while explicitly accounting for gaze shifts [Sinz2018, NeurIPS], that we will include in our final references. However, the proposed approach is based on knowing fixation locations during training/testing and is restricted to predicting V1 responses.

**[R1 - "neural response is dominated by sensory signals at attended locations - what is the novelty?"]** We understand the reviewer's concern. While it is known that neural responses are dominated by sensory signals at attended locations, we here demonstrate that this property can indeed be leveraged in the neural response prediction task with the dual goal of (a) improving response prediction in later stages of the visual pathway and (b) learning attention policies employed by humans while viewing naturalistic scenes without employing any eye-tracking data.

**[R1 - "covert attention maps?"]** Since the learned attention model does not employ fixation data at all, we believe the model is equally well capable of modeling covert attention. In fact, the discrepancy between the predicted saliency and human fixation maps could partially also be explained by covert mechanisms of attention, although this speculation may be pre-mature given current analysis. Since the learned attention model is only trained to maximize neural response prediction accuracy, in principle, the attention sub-network therein should learn to focus on locations within the stimulus insofar as they dominate neural representations.

**[R4 - scale of heatmaps and code]** We apologize for the missing colorbars and will include them in the final version. We will also provide link to the code and trained models in the final version.

[Meta-Review · NeurIPS 2020]

This paper was positively reviewed by all reviewers and even more so after the rebuttal. All reviewers agree with selecting it for oral presentation.